# SYNTHESISING AUDIO ADVERSARIAL EXAMPLES FOR AUTOMATIC SPEECH RECOGNITION

## ABSTRACT

Adversarial examples in automatic speech recognition (ASR) are naturally sounded by humans *yet* capable of fooling well trained ASR models to transcribe incorrectly. Existing audio adversarial examples are typically constructed by adding constrained perturbations on benign audio inputs. Such attacks are therefore generated with an audio dependent assumption. For the first time, we propose the Speech Synthesising based Attack (SSA), a novel threat model that constructs audio adversarial examples entirely from scratch, i.e., without depending on any existing audio) to fool cutting-edge ASR models. To this end, we introduce a conditional variational auto-encoder (CVAE) as the speech synthesiser. Meanwhile, an adaptive sign gradient descent algorithm is proposed to solve the adversarial audio synthesis task. Experiments on three datasets (i.e., Audio Mnist, Common Voice, and Librispeech) show that our method could synthesise audio adversarial examples that are naturally sounded but misleading the start-of-the-art ASR models. The project webpage containing generated audio demos is at `https://sites.google.com/view/ssa-asr/home`.

## 1 INTRODUCTION

Deep neural network (DNN) based models have documented many success stories in various domains, such as reinforcement learning (Silver et al., 2017), image classification (Deng et al., 2009), and automatic speech recognition (ASR) (Chan et al., 2016). However, DNN models are found to be vulnerable to adversarial attacks (Goodfellow et al., 2015), viz., the slightly perturbed input would cause severe errors or performance drops of well trained DNN models. This paper mainly focuses on the ASR domain, where many voice assistant systems could be hijacked or controlled by the audio adversarial examples constructed by an attacker.

To investigate the threat of audio adversarial examples, many different approaches (Carlini & Wagner, 2018; Yuan et al., 2018; Yang et al., 2019; Qin et al., 2019) have been developed. In general, existing approaches assume that the semantics (for human beings) of adversarial audio can be preserved as long as the distance between the adversarial audio and the benign audio is restricted properly. In particular, a shared design principle is to add constrained adversarial perturbations (i.e., as imperceptible as possible) on benign audios yet with the goal of fooling ASR models significantly. For instance, Qin et al. (2019) introduced a psychoacoustic rule of auditory masking to only add perturbation on a benign audio where the noise is hard to be heard. Therefore, existing approaches can be deemed as *audio dependent attack* (ADA), viz., the audio adversarial examples have to be constructed depending on some benign audios as shown in Figure 1 (a). However, in real cases, the human speaker and/or the benign audio may not be available or accessible. Moreover, ADA relies on an imperceptible perturbation principle, viz., the added perturbation must be restricted enough to avoid being perceived by human beings.

In contrast, this paper proposes the audio independent attack (AIA) as shown in Figure 1 (b) that sheds light on a more general principle of adversarial attacks, viz., ***any** audio that deceives ASR models yet fails to deceive human beings would cause security issues in speech recognition*. Alternatively stated, an adversarial audio does not have to be constructed based on an existing benign audio as ADA does. Our AIA thus enables a novel threat model that constructs audio adversarial examples completely from scratch instead of adding perturbations on existing benign audios. Particularly, mounted on recent advances in neural speech synthesis (Tan et al., 2021), we can directly

Figure 1: Audio dependent attack *versus* audio independent attack.

synthesise the adversarial audio that conserves the desired semantic content but deceives the ASR model to predict incorrect or even targeted transcriptions.

With this goal in mind, we propose the Speech Synthesising based Attack (SSA) as shown in Figure 2, where a conditional variational autoencoder CVAE is incorporated to synthesise audio waveform $x$ that is connected to an ASR model thereafter. The basic philosophy of SSA is: the audio style vector $z$ in CVAE controls the pitches and rhythms of synthesised waveforms that may cause the ASR model to transcribe incorrectly or even as a target; thus some particular synthesised waveforms would become adversarial examples. In particular, following (Kim et al., 2021), we first train CVAE by adopting a variational inference augmented with normalizing flows and an adversarial training process, so as to synthesise natural sounding audios. Once the CVAE is well trained, given a sequence of conditional text that represents the ground truth audio semantics, we can optimize $z$ to get a particular $z*$ that can deceive an advanced ASR model through a reg-

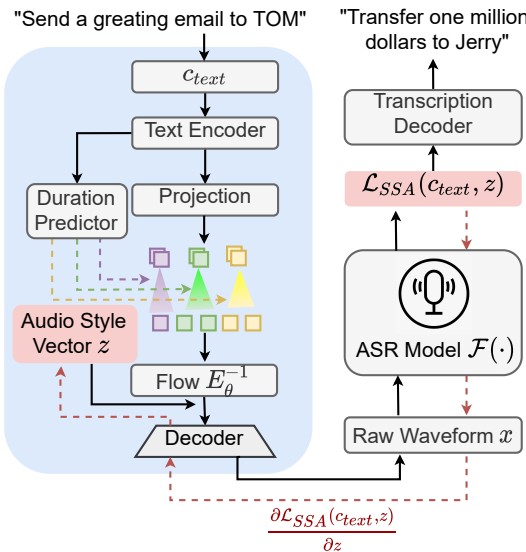

Figure 2: The general structure of SSA.

ularized connectionist temporal classification loss. In this regard, we formulate the adversarial example synthesising task as a gradient sign based optimization problem. More importantly, to solve the optimization efficiently, we design an adaptive learning rate decay based on the annealing mechanism, which can automatically adjust the step size during optimization. Our contributions are summarized as:

- For the first time, we propose an audio independent adversarial attack as a novel threat model in ASR, which constructs adversarial example completely from scratch without depending on any existing speaker/audio. This sheds light on a more general principle of adversarial attacks, viz., *any* audio that deceives ASR models yet fails to deceive human beings would cause security issues in speech recognition.

- We develop SSA that adopts CVAE as the speech synthesiser. To efficiently synthesise audio adversarial examples, we establish an adaptive sign gradient descent algorithm via designing an annealing mechanism inspired learning rate decay.

- Extensive experiments across three datasets (i.e., Audio Mnist (Becker et al., 2018), Common Voice (Ardila et al., 2019), and Librispeech (Panayotov et al., 2015)) based on DeepSpeech model (Amodei et al., 2016) show that our SSA can effectively generate audio adversarial examples.

## 2 RELATED WORK

In ASR adversarial attacks, Carlini & Wagner (2018) were among the first ones to showcase that slight perturbations on audio can easily fool a start-of-the-art ASR model to transcribe the perturbed audio into any target sentence. Later on, Yuan et al. (2018) demonstrated that such adversarial audio perturbation can be embedded into a song. Yang et al. (2019) analyzed that the temporal dependency could promote the discriminative power against adversarial examples in ASR. Moreover, Khare et al.

(2019) and Taori et al. (2019) found that similar audio adversarial perturbations could be generated using black-box optimization algorithms (e.g., genetic algorithm (Mitchell, 1998)). To generate the audio adversarial examples faster and improve the attack efficiency, Liu et al. (2020) proposed to adaptively rectify the weights of audio perturbations on different positions. To make the audio perturbation more imperceptible, Qin et al. (2019) introduced a psychoacoustic principle of auditory masking to smartly add perturbations on frames where noises are hard to be perceived. Xie et al. (2021) utilized generative adversarial network (GAN) to generate adversarial examples on speech domain, where GAN is to learn the distribution of predefined adversarial perturbations. Thus the adversarial examples are still generated depending on the benign audios.

In summary, previous adversarial attacks on ASR are audio-dependent, viz., the audio adversarial examples must be generated based on corresponding benign audios. Although there are studies, e.g., (Song et al., 2018; Wang et al., 2019), on generating adversarial examples from scratch, but most of them focus on the image domain. Carlini & Wagner (2018) studied an audio attack that starts from non-speech (e.g., a piece of classic music), while existing audios are stilled required to mount perturbations on. Roy et al. (2018) and Zhang et al. (2017) proposed to modulate voice commands on ultrasonic carriers (e.g., frequency $> 20\ kHz$) to achieve inaudibility, where the scenario is an adversary that stands on the road and silently controls the voice command assistant systems. This therefore is different from our scope of generating natural sounding adversarial audios. To the best our knowledge, our work is the first attempt to generate audio adversarial examples from scratch without utilizing benign audios in ASR.

## 3  BACKGROUND

**Automatic Speech Recognition (ASR).** In this paper, we focus on the speech-to-text tasks that are based on neural ASR models. Following previous studies (Esmaeilpour et al., 2021; Liu et al., 2020; Yakura & Sakuma, 2019; Carlini & Wagner, 2018), we stick our attacks on the **Deep-Speech** (Amodei et al., 2016), a state-of-the-art ASR model based on Connectionist Temporal Classification (CTC) method (Graves et al., 2006). DeepSpeech uses CTC as the input aduios and the corresponding transcriptions are unaligned, viz., the exact position of each word in the audio sample is unknown. To enable efficient supervised training, a transcription is first enumerated to obtain all alignments. Hence, the CTC loss is minimized to maximize the probabilities over all alignments.

**CTC Loss.** Let $\mathcal{X}$ be the audio input domain and $\mathcal{Y}$ be the text output domain with dimension $|\mathcal{Y}|$. The ASR model is donoted as $\mathcal{F} : \mathcal{X}^N \rightarrow \mathcal{Y}^{N \cdot |\mathcal{Y}|}$, which takes a $N$ frames $x \in \mathcal{X}$ as input and outputs a probability distribution $\mathcal{F}(x)$ over the output domain. Given $y$ as a phrase (i.e., a sequence of characters), we define a token sequence $\pi$ being reducible to $y$ if the two operations, namely, removing sequentially duplicated tokens, and deleting all blank tokens on $\pi$ could produce $y$. We further denote $\pi$ as an alignment of $y$ if $\pi$ can be reduced to $y$ and the length of $\pi$ equals to the length of prediction $\mathcal{F}(x)$. Let $\Phi(y)$ be the alignment set obtained from targeted transcription $y$ using dynamic programming (Graves et al., 2006). Accordingly, the CTC loss can be formulated as,

$$\mathcal{L}_{ctc}(x, y) = -log \left[ \sum\nolimits_{\pi \in \Phi(y)} \prod\nolimits_i^N \mathcal{F}(x)_{\pi_i}^i \right], \tag{1}$$

where $\mathcal{F}(x)_{\pi_i}^i$ is the probability of the token $\pi_i$ on the $i_{th}$ frame. To get the transcription in inference, a decoder $\mathcal{D}$ (e.g., greedy decoding or beam search decoding) is required. Thereby, if an ASR model is well trained, the transcription would satisfy $y = \mathcal{D}(\mathcal{F}(x)) = f(x)$, where $f(\cdot)$ is a merged denotation of $\mathcal{F}$ and $\mathcal{D}$.

**White-Box Assumption.** Following most of previous studies, we use the similar white-box assumption, viz., the parameter and structure of ASR model is known. Investigating the black-box attack is another direction. Moreover, there are many ways (Oh et al., 2019; Zanella-Béguelin et al., 2021; Qin et al., 2019) to convert a black-box model to a white-box one.

**Targeted Attack.** Compared with the untargeted attack that only maximizes the word error rate (WER), the targeted attack is a more challenging task since it requires not only the audio perturbation imperceptible, but also the adversarial audio being transcribed to a specified target phrase. Our main focus is on the targeted scenario. Without specification, the adversarial example in following sections refers to the targeted one.

**Speech Synthesise (End-To-End Text-To-Speech).** Text-to-speech (TTS) model synthesises waveforms given text phrases as semantic contents. For efficiently synthesising adversarial audios in our SSA, we turn to the end-to-end TTS model that could easily utilize modern parallel processors for faster synthesis speed. In particular, we utilize a conditional variational autoencoder (Kim et al., 2021) based TTS model to generate natural sounding audios, which is explained in Section 4.2.

# 4 METHODS

## 4.1 PROBLEM SETTINGS

Given a well trained ASR model $f(\cdot)$, the objective of adversarial attack is to construct an audio waveform $x \in \mathcal{X}$ that is naturally sounded yet able to deceive $f(\cdot)$ in predicting incorrect/targeted transcriptions. Suppose $o : \mathcal{X} \rightarrow \mathcal{Y}$ is an oracle that takes an audio waveform $x$ as input and outputs the *ground truth* transcription $y_o = o(x)$, where $\mathcal{Y}$ is the set of all text transcriptions under consideration. Moreover, compared to the untargeted attack that only introduces spelling errors, we focus on the more challenging targeted attack, viz., $f(x) = y$ where $y \in \mathcal{Y}$ is the expected target transcription by an attacker.

In previous studies, the adversarial audio $x$ is constructed by adding perturbation $\delta$ on a benign audio $x_o$, viz., $x = x_o + \delta$; thus being dependent on $x_o$. Mounted on these notations, we give a formal definition of the previous audio dependent attack as follows.

**Definition 1** (Audio Dependent Attack - ADA). Given a benign audio $x_o$ and its oracle transcription $y_o = o(x_o)$, the corresponding adversarial example can be defined as any audio $x$, viz., $x \in \mathcal{A}_\delta \triangleq \{x_o + \delta \in \mathcal{X} | \exists \delta, \underbrace{\mathcal{M}(\delta) \leq \epsilon} \cap \underbrace{o(x_o + \delta) = y_o = o(x_o)} \cap \underbrace{f(x_o + \delta) = y_t} \cap \underbrace{f(x_o) \neq y_t} \cap \underbrace{y_o \neq y_t}\}$, where $\mathcal{M}(\cdot)$ is a distance measurement (e.g., matrix norm); $\epsilon$ is a small positive constant; and $y_t$ indicates the targeted transcription of an attacker.

From Definition 1 the adversarial audio $x$ is directly built on a benign audio $x_o$. Moreover, to guarantee $x$ being acoustically realistic or natural, the efforts mainly focus on forcing $\mathcal{M}(\delta) \leq \epsilon$ with the goal of restricting $x$ to be close to $x_o$. However, in some cases, the benign audio $x_o$ may not be available. For instance, an attacker want to deceive a voice commander when no human speaking happens nearby. Moreover, ADA relies on an imperceptible perturbation principle, viz., the added perturbation $\delta$ must be small enough to avoid being perceived by human beings. In contrast, a more general scenario would be audio independent attack that is defined as follows.

**Definition 2** (Audio Independent Attack - AIA). Given a conditional text (i.e., $y_o$), an audio independent adversarial example can be any element from $\mathcal{A}_a \triangleq \{x \in \mathcal{X} | \underbrace{o(x) = y_o} \cap \underbrace{f(x) = y_t} \cap \underbrace{y_o \neq y_t}\}$, where $o(x) = y_o$ indicates that the synthesised audio $x$ correctly conveys its semantic content $y_o$; and $f(x) = y_t$ means the ASR model $f(\cdot)$ is successfully fooled to output the targeted transcription $y_t$.

Such AIA sheds light on a more general principle of adversarial attacks, viz., *any audio that deceives ASR models yet fails to deceive humans would cause security issues in speech recognition*. In other words, the adversarial audios are not necessarily constructed via adding perturbations. Instead, they can be directly synthesized with the goal of preserving the desired semantic content (i.e. $o(x) = y_o$), while simultaneously deceiving the ASR model to predict incorrect or even targeted transcriptions (i.e. $f(x) = y_t$). This motivates us to leverage the powerful generative model in TTS area to construct such adversarial audio $x$. Next, we will introduce the speech synthesising based attack.

## 4.2 SPEECH SYNTHESISING BASED ATTACK

The overall structure of the speech synthesis based attack (SSA) is depicted in Figure 2.

**Conditional Variational Autoencoder based Speech Synthesis.** The key for synthesising natural sounding adversarial audio is to model the TTS mapping. In practice, we can select different types of TTS models (Tan et al., 2021) to generate natural sounding audios. We choose the recent conditional variational autoencoder (CVAE) based TTS model (Kim et al., 2021) due to its rich variety in audio generation, viz., a text input can be spoken in multiple ways with different pitches and rhythms.

---

**Algorithm 1:** SSA Algorithm

---

**Input:** $y_o$ – the ground truth text; $y_t$ – the target for attack; $\mathcal{G}(\cdot)$ – the generative model for speech
synthesise; $\mathcal{F}(\cdot)$ – the ASR model; $\alpha_0$ – the initial learning rate; $d_\alpha$ – the learning rate decay ratio;
$\lambda$ – the weight of $\mathcal{L}_{reg}(z)$; $I_m$ – the maximum iteration number; $D_{ls}(\cdot)$ – Levenshtein distance.

1   Initialization: audio style vector $z \sim \mathcal{N}(\mathbf{0}, \mathbf{1})$; the patience $p = 0$; record best loss $\mathcal{L}* = +\infty$;
2   **for** $i \in \mathbf{N}_+ \cap i < I_m$ **do**
3     Calculate the SSA loss $\mathcal{L}_{ssa}(z, y_o, y_t)$ in Eq. (3);
4     **if** $\mathcal{L}_{ssa}(z, y_o, y_t) <= \mathcal{L}*$ **then**
5       $\lfloor$   $\mathcal{L}* = \mathcal{L}_{ssa}(z, y_o, y_t), p = 0$
6     **else**
7       $\lfloor$   $p \leftarrow p + 1$
8     **if** $p >= p_m$ **then**
9       $\lfloor$   $\alpha \leftarrow d_\alpha \cdot \alpha, p = 0$
10    Do back-propogation to get $\frac{\partial \mathcal{L}_{ssa}(z, y_o, y_t)}{\partial z}$ and update $z$ following Eq. (5);
11    **if** $D_{ls}(f(\mathcal{G}(z, y_o)), y_t) == 0$ **then**
12      $\lfloor$   **Return** the successful audio adversarial example $x* = \mathcal{G}(z, y_o)$

13   **Return** the current best audio adversarial example $x = \mathcal{G}(z, y_o)$

---

This accordingly enlarges the sample space, and enhances the possibility of constructing successful adversarial examples. In specific, such CVAE based TTS model (Kim et al., 2021) is based on three components: 1) a conditional VAE formulation; 2) an alignment estimation derived from variational inference; 3) an adversarial training for improving the synthesis quality. The architecture of the CVAE model in inference is shown in Figure 2, where the corresponding training loss and settings can refer to (Kim et al., 2021).

In particular, given a conditional text $c_{text} = y_o$ (where $y_o$ conveys the ground truth semantic content), the TTS model aims to build the mapping $\mathcal{G}(\cdot)$, viz.,

$$x = \mathcal{G}(z, c_{text}), z \sim \mathcal{N}(\mathbf{0}, \mathbf{1}), \tag{2}$$

where $z$ is a normally distributed vector that *controls the audio styles with different pitches and rhythms*. Therefore, with different $z$, there are different ways to speak the content in $c_{text}$. This aligns to the fact, viz., even human beings always pronounce same words differently from time to time. Note that the audio $x$ generated by CVAE model $\mathcal{G}(\cdot)$ in (Kim et al., 2021) has been tested to be natural sounding using the mean opinion score obtained from Amazon Mechanical Turk (www.mturk.com). The following goal thus becomes figuring out a particular $z$ that can fool an ASR model $\mathcal{F}(\cdot)$.

**Speech Synthesising based Attack (SSA) Formulation.** We focus on the targeted attack, viz., deceiving the ASR model $\mathcal{F}(\cdot)$ to predict a target phrase $y_t$. Given the speech synthesiser $\mathcal{G}(z, c_{text})$ in Eq. (2), our SSA is formulated as finding a particular $z*$ during synthesising audio $x$ that can enable the targeted attack. To this end, a loss function $\mathcal{L}_{ssa}$ is designed to construct natural sounding yet fooling enough $x$. Accordingly, the optimization of $z$ can be defined as

$$z* = \arg\min_z \mathcal{L}_{ssa}(z, y_o, y_t) = \arg\min_z \mathcal{L}_{ctc}(\mathcal{G}(z, y_o), y_t) + \lambda \mathcal{L}_{reg}(z), \tag{3}$$

where $y_o$ is the phrase that the CVAE model wants to synthesise; $y_t$ is the targeted phrase that the ASR model is fooled to predict; and $\mathcal{L}_{reg}(z)$ is the regularization loss controlled by $\lambda$. To be specific, $\mathcal{L}_{reg}(z)$ is designed to boost the generated audio to be naturally sounded. Based on (Kim et al., 2021), $z$ is sampled from a normal distribution. To preserve this property, we design it as:

$$\mathcal{L}_{reg}(z) = \phi(\mu(z)) + \phi(\delta(z) - 1), \tag{4}$$

where $\phi(\cdot)$ is an absolute value function; $\mu(z)$ and $\delta(z)$ are the mean and variance of $z$, respectively. To detemine whether the generated audio $x = \mathcal{G}(z, y_o)$ can enable a successful targeted attack (i.e., $f(x) == y_t$), we involve the Levenshtein distance (Yujian & Bo, 2007). During the optimization of $\mathcal{L}_{ssa}(z, y_o, y_t)$, if $D_{ls}(f(\mathcal{G}(z, y_o)), y_t) = 0$, the targeted attack is successful.

**Adaptive Sign Gradient Descent.** To solve the optimization problem in Eq. (3), we propose an adaptive sign gradient descent algorithm whereby an adaptive learning rate decay mechanism is

designed in the sign gradient descent optimization. Typically, the sign gradient descent is frequently adopted in adversarial attack algorithms, e.g., fast gradient sign method (Goodfellow et al., 2015) and projected gradient descent (Madry et al., 2018). Moreover, Balles & Hennig (2018) theoretically proved the benefits of using gradient sign as the optimization direction. In our SSA, the sign gradient descent based optimization is given by,

$$z \leftarrow z + \alpha \frac{\partial \mathcal{L}_{ssa}(z, y_o, y_t)}{\partial z}, \tag{5}$$

where $\alpha$ is the learning rate, which significantly impacts the convergence of the SSA optimization. Inspired by the heuristics of annealing (Bertsimas & Tsitsiklis, 1993), we carefully design an adaptive learning rate decay mechanism as below.

**Definition 3** (Annealing based adaptive learning rate decay). For every $p_m$ steps where the attack loss $\mathcal{L}_{ssa}(z, y_o, y_t)$ has no improvement, the learning rate $\alpha$ will decay as $\alpha := d_\alpha \cdot \alpha$, where $d_\alpha \in [0, 1]$ is the decay ratio.

Definition 3 tells that if a specific $\alpha$ gets stuck for particular $p_m$ steps with regard to $\mathcal{L}_{ssa}(\cdot)$, $\alpha$ should decay to perform a more local search. The overall process of SSA is shown in Algorithm 1.

## 5 EXPERIMENTS AND RESULTS

### 5.1 EXPERIMENT SETTINGS

**Datasets.** In our experiments[1], we use three datasets, i.e., Audio Mnist (Becker et al., 2018), Common Voice (Ardila et al., 2019), and Librispeech (Panayotov et al., 2015). However, different from previous studies using both the waveform and text label, we only utilize the text information due to the audio independent property of our SSA. In particular, the Audio Mnist contains the text digits (i.e., from "ZERO" to "NINE"). When building the targeted attack pairs, we choose one text digit (e.g., "ZERO") as $y_o$ and enumerate the rest digits (i.e., from "ONE" to "NINE") as its attack target $y_t$. For Common Voice and Librispeech, we randomly sample 1000 text labels first. Those sampled texts are filtered and clustered based on their text length. Thereafter, 100 texts are sampled out from the filtered 1000 samples as the candidates of conditional text $y_o$. The corresponding target text $y_t$ is sampled from their matching clusters. The length comparison between $y_o$ and $y_t$ on the two datasets are shown in Figures (10-11) in the appendix, where we notice that $y_o$ and $y_t$ are aligned with a similar length. More analysis can refer to appendix 7.1. Our constructed datasets[2] are released to benefit future research on speech synthesise/generative model related attacks.

**Model Settings.** In our SSA paradigm, the CVAE model and DeepSpeech are utilized as speech synthesiser and speech recogniser, respectively. The CVAE model follows the setting of VITS (Kim et al., 2021) during inference. In particular, we utilize the CVAE model trained on VCTK dataset (Veaux et al., 2017), which can be obtained from VITS[3]. The standard deviation of the input noise for the stochastic duration predictor is set to be 0, thus making it a deterministic one. In addition, a scaling factor for $z$ is applied. The DeepSpeech model follows the Pytorch implementation in the adversarial robustness toolbox (Nicolae et al., 2018).

**Evaluation Metrics.** Two evaluation metrics, i.e., the word error rate (WER) and success rate (SR), are adopted under the targeted attack setting. In particular, they are defined as,

$$WER = \frac{S + D + I}{N_w}, \quad SR = \frac{N_s}{N_a} \tag{6}$$

where $S$, $D$ and $I$ indicate the number of subsitutions, deletions and insertions of words respectively; $N_w$ is the total number of words; $N_s$ is the number of successful adversarial example (i.e., $f(x) = y_t$); and $N_a$ is the total number of audios synthesised. Note that $SR$ is same as the sentence-level accuracy that is used in previous studies (Qin et al., 2019). In general, larger values of WER and SR indicate a stronger attack algorithm.

**SSA Optimization Settings.** The learning rate $\alpha$ decays in updating $z$ as highlighted in Definition 3 and Algorithm 1. In particular, $\alpha_0$ (i.e., initialization of learning rate) is searched in the range of

---

[1] Our code will be released upon acceptance.

[2] https://drive.google.com/file/d/1EHXRlWrlMXr6qu8WYtjt8-pRzw-VfCau/view?usp=sharing

[3] https://github.com/jaywalnut310/vits

Table 1: Targeted results of SSA on the three datasets and comparisons with baselines.

| | Attack algorithms | WER (%) | SR (%) |
|---|---|---|---|
| **Baselines** | DeepSpeech (No Attack) | 7.55 | NA |
| | C&W (Carlini & Wagner, 2018) | $78.94 \pm 2.01$ | $30.74 \pm 3.16$ |
| | Y&S (Yakura & Sakuma, 2019) | $80.28 \pm 3.14$ | $35.49 \pm 0.28$ |
| | GAA (Taori et al., 2019) | $65.80 \pm 2.55$ | $48.35 \pm 3.38$ |
| | MOOA (Khare et al., 2019) | $68.06 \pm 2.71$ | $47.01 \pm 1.42$ |
| | Metamorph (Chen et al., 2020) | $72.48 \pm 1.06$ | $45.84 \pm 4.71$ |
| | CIPMA (Esmaeilpour et al., 2021) | $88.19 \pm 3.15$ | $21.69 \pm 3.09$ |
| **SSA** | SSA-Audio Mnist | $100.00 \pm 0.00$ | $\mathbf{100.00 \pm 0.00}$ |
| | SSA-Common Voice | $\mathbf{106.27 \pm 18.03}$ | $96.00 \pm 8.00$ |
| | SSA-Librispeech | $100.71 \pm 13.21$ | $83.19 \pm 16.33$ |
| | **SSA-Average (Common Voice & Librispeech)** | $103.49 \pm 16.05$ | $89.60 \pm 14.37$ |

Table 2: The MOS comparison between original and SSA synthesised audios.

| Type of audios | MOS |
|---|---|
| Before attack (original synthesised audios) | $4.09 \pm 0.10$ |
| After attack (SSA synthesised audios) | $3.39 \pm 0.29$ |

$[0.01, 0.09]$ stepped by 0.01; the patience $p_m$ ranges from 50 to 400 stepped by 50; the learning rate decay ratio $d_\alpha$ is searched in $\{0, 5, 0.6, 0.7\}$; moreover, the regularization weight $\lambda$ is searched in $\{0, 50, 100, 150, 200\}$. The maximum iteration number $I_m$ is 8000, where any case without finding a successful attack within $I_m$ budget is deemed as failed.

## 5.2 TARGETED ATTACK PERFORMANCE

We first show that our proposed SSA is capable of efficiently constructing audio adversarial examples, viz., significantly outperforming existing baselines in Table 1. In particular, the baselines include DeepSpeech[4], C&W attack (Carlini & Wagner, 2018), Y&S attack (Yakura & Sakuma, 2019), GAA attack (Taori et al., 2019), MOOA attach (Khare et al., 2019), Metamorph attack (Chen et al., 2020), and CIPMA attack (Esmaeilpour et al., 2021). The results from the baselines are reported in (Esmaeilpour et al., 2021), which are averaged results over Common Voice and Librispeech. Our SSA is evaluated on three datasets (i.e., Audio Mnist, Common Voice, and Librispeech), where an averaged result over Common Voice and Librispeech is calculated for a fair comparison. All the results of our SSAs are based on targeted attack settings.

In general, we can see that our SSA achieves remarkably better performance on both WER and SR. Specifically, (1) SSA-Average achieves $103.49\%$ and $89.60\%$ regarding WER and SR, respectively, which outperforms the best results obtained by CIPMA on WER (i.e. $88.19\%$) and GAA on SR (i.e. $48.35\%$); (2) both SSA-Common Voice and SSA-Librispeech dramatically outperform the best results obtained from the baselines as well. Especially for SSA-Common Voice, both of its WER (i.e. $106.27\%$) and SR (i.e. $96.00\%$) are the best across all comparisons; (3) SSA-Librispeech behaves worse than SSA-Common Voice with regards to WER and SR, which is mainly attributed to the difference of conditional text lengths between the two datasets; and (4) our proposed SSA realizes $100\%$ w.r.t. both WER and SR on Audio Mnist, which can serve as a strong baseline for future study on the audio adversarial attack. We suggest the reader to listen to our synthesised adversarial audios that are available at our webpage[5].

## 5.3 QUALITY EVALUATION OF SYNTHESISED AUDIOS

Although the regularization loss $\mathcal{L}_{reg}(z)$ in our SSA is designed to force our synthesised audios to be independently and identically distributed w.r.t. the audio generated in VITS [1], we still find some synthesised audios not natural sounding enough. Therefore, we further evaluate the quality of synthesised adversarial audios by mean opinion score (MOS) tests. Specifically, we invited 20 participants to rate on 50 sample pairs, where each sample pair includes an original synthesised audio by CVAE and its corresponding synthesised adversarial audio optimized by our SSA. Each participant will listen to these 100 audio samples that are randomly shuffled. The naturalness rating

---

[4] https://github.com/Picovoice/speech-to-text-benchmark#mozilla-deepspeech
[5] https://sites.google.com/view/ssa-asr/home

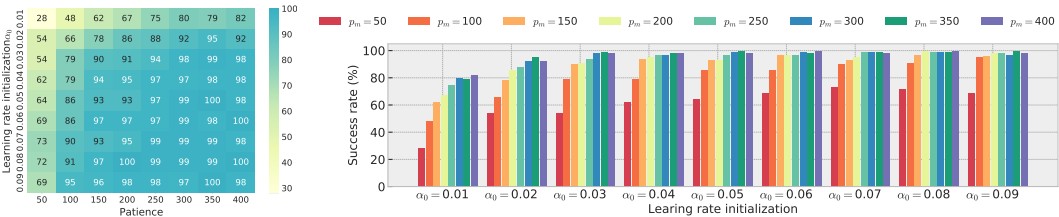

Figure 3: The results of SR with different $\alpha_0$ and $p_m$ on Audio Mnist.

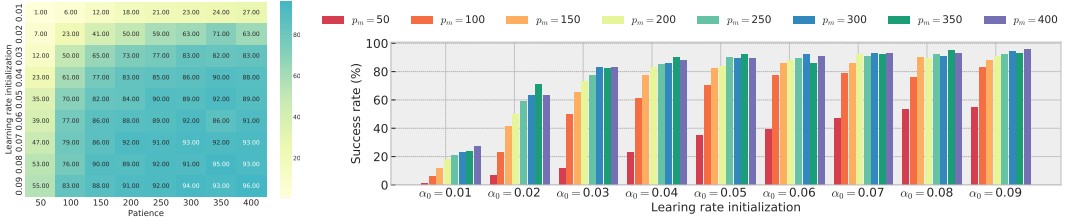

Figure 4: The results of SR with different $\alpha_0$ and $p_m$ on Common Voice.

score is scaled from 1 to 5. The MOS results are shown in Table 2, where we can observe a slightly worse MOS score of our SSA synthesised audios (i.e., $3.39 \pm 0.29$) compared with that of original synthesised ones (i.e., $4.09 \pm 0.10$). This again indicates that there exists distortions in the synthesised audios by SSA. However, we further note that the difference of MOS scores between the two types of synthesised audios is not significant, which indicates that the distortions are still subjectively acceptable. Future works can focus on how to eliminate such distortions in the adversarial audio generation, such as redesigning the regularization loss $\mathcal{L}_{reg}(z)$ in Eq (3).

## 5.4 HYPERPARAMETER ANALYSIS ON THE ADAPTIVE SIGN GRADIENT DECENT

In the adaptive sign gradient decent, the initial learning rate $\alpha_0$ and the patience $p_m$ are two important hyperparameters. Their impacts regarding SR on Audio Mnist and Common Voice datasets are displayed in Figure 3 and 4, respectively. Moreover, we also analyze their impacts on Common Voice w.r.t. WER. Several interesting findings are noted as below.

**Analysis on SR. (1)** Generally, the SR first climbs up as $\alpha_0$ increases and then keeps stable with further increase of $\alpha_0$. For instance, a stable SR is achieve with $\alpha = 0.03$ on Audio Mnist. Similar trends are also held with $p_m$. **(2)** On both Audio Mnist and Common Voice, $p_m$ needs to be set as a mild value (e.g., $p_m \in \{300, 350\}$) in order to obtain a promising SR. This indicates that if the learning rate $\alpha$ decays too fast, the sign gradient descent optimization in Algorithm 1 may get stuck. **(3)** Compared the results in Figures (3-4), the SR from Common Voice is usually smaller than that from Audio Mnist even with the same settings on $\alpha_0$ and $p_m$. This is mainly because the audio synthesised in Common Voice is clearly longer than that from Audio Mnist.

**Analysis on WER.** Figure 5 shows the WER with different settings of $\alpha_0$ and $p_m$. As a whole, the WER first goes up with the increasing of both parameters, while becomes less sensitive as they continue to increase. For example, comparing across different lines (i.e., with different $\alpha_0$), we can only ob-

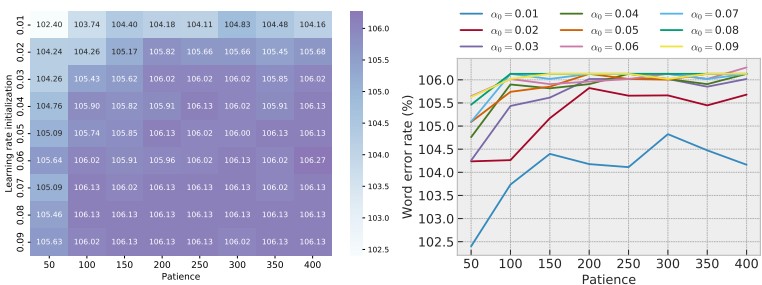

Figure 5: WER with different $\alpha_0$ and $p_m$ on Common Voice.

serve a slight change on WER with $\alpha_0 \geq 0.04$. This leads to a similar conclusion with the analysis on SR, viz., proper settings of $\alpha_0$ and $p_m$ can easily synthesise harmful adversarial audios via SSA.

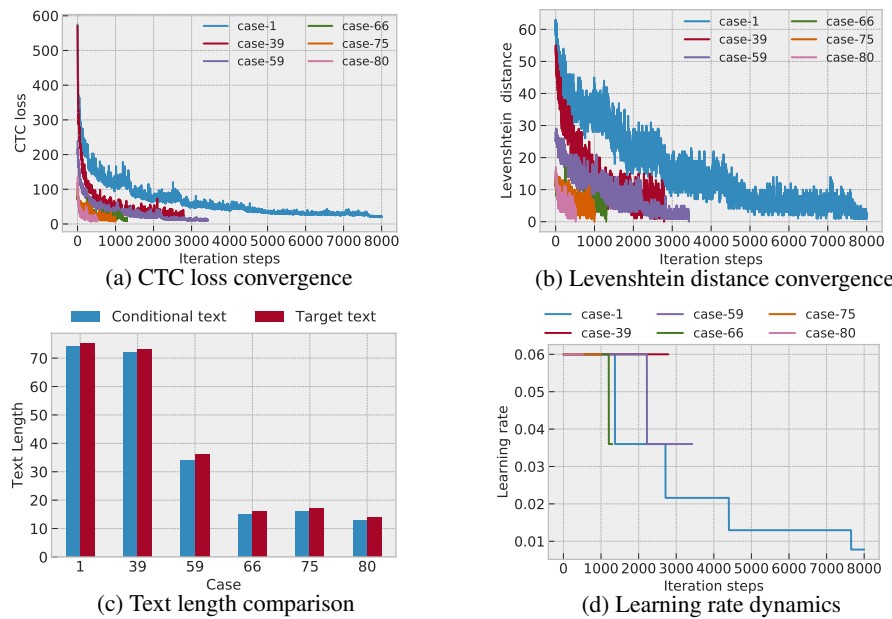

(a) CTC loss convergence

(b) Levenshtein distance convergence

(c) Text length comparison

(d) Learning rate dynamics

Figure 6: Convergence analyses on selected representative cases on Common Voice.

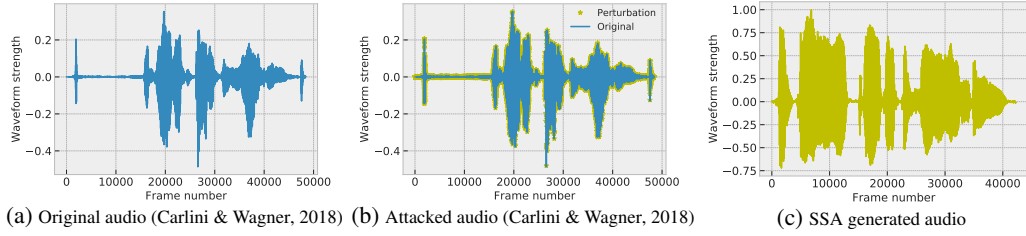

(a) Original audio (Carlini & Wagner, 2018) (b) Attacked audio (Carlini & Wagner, 2018)

(c) SSA generated audio

Figure 7: The comparison on waveforms between the audio dependent attack and our SSA.

## 5.5 CONVERGENCE ANALYSIS

One natural question is, *given a particular setting on $\alpha_0$ and $p_m$, how does the process of adaptive sign gradient descent look like?* We choose the best setting according to the results in Figure 4, viz., $\alpha_0 = 0.06$ and $p_m = 400$, and select six representative evaluated cases to show the convergence of the CTC loss, Levenshtein distance on Common Voice in Figures 6(a-b), respectively. In addition, to assist the analysis, the correspond text length comparison and learning rate decays are provided in Figures 6(c-d). Due to space limitation, the overall convergence process across 100 samples related to CTC loss and Levenshtein distance and learning rate decays are respectively shown in Figures (12-14) in the appendix. Some interesting observations are noted as follows.

First, from Figure 6(a) and Figure 12, we can see that the CTC loss of most cases quickly converges to a small value close to 0. The convergence speed is especially fast at the beginning of the optimization. Such a phenomenon is also observed in other studies (Amodei et al., 2016) related to CTC loss based training. Second, in Figure 6(b), the Levenshtein distance on most cases except case 1 converges to 0, indicating a successful targeted attack. Third, on some cases, e.g., case 1 in Figure 6(a), although the convergence curve does not stop before reaching the maximum iteration step 8000, it can be deemed as an approximately successful attack as reflected by the small CTC loss in Figure 6(a) and Levenshtein distance in Figure 6(b). Fourth, the additional text length comparison in Figure 6(c) shows that the text length generally reflects the difficulty of generating a successful attack, viz., a longer text usually requires more iterations in both CTC loss and Levenshtein distance convergences. Lastly, the learning rate dynamics in Figure 6(d) also showcase that for a harder problem (i.e., with a larger text length), the learning rate decays more times to exploit a solution close to a successful attack. More results and analyses can refer to Appendix 7.2.

## 5.6 WAVEFORM PATTERN ANALYSIS

To showcase that our SSA is a more general attack compared with audio dependent attacks, we further analyse the waveform pattern of both attacks as shown in Figure 7. In particular, Figure 7(a) and (b) respectively depict the original audio and the corresponding adversarially perturbed audio, where we can easily observe that the attacked audio needs to be restricted to only add minor perturbations. In contrast, the adversarial audio constructed by our SSA as shown in Figure 7(c) is free of such restriction, viz., the waveform can be significantly different. In sum, our SSA can be deemed as an audio independent attack, which brings in more threat to ASR models in the wild. Further analyses towards the audio style vector $z$ before and after attack of SSA are shown in Appendix 7.3. In addition, we also evaluate the tranferability of the adversarial audios w.r.t another ASR model (i.e., ESPnet (Watanabe et al., 2018)) as shown in Appendix 7.4.

## 6 CONCLUSION

This paper investigates the audio adversarial attack for ASR models. Existing attack algorithms are based on an audio dependent assumption, viz., adding constrained perturbations on benign audio inputs. In contrast, we propose SSA, a novel threat model that constructs audio adversarial examples entirely from scratch, viz, without depending on any existing audio to fool cutting-edge ASR models. To this end, we propose to use a conditional variational auto-encoder (CVAE) as the speech synthesiser. Accordingly, the adversarial audio synthesising task is formulated as an optimization problem via searching in the hidden space of CVAE. Meanwhile, an adaptive sign gradient descent algorithm is further devised to solve the SSA optimization problem. Experiments on three datasets show that our proposed SSA can synthesise audios that are naturally sounded but deceive start-of-the-art ASR models. In our experiments, we also find that some synthesised adversarial audios do not sound as natural as those without any manipulation on $z$, which thus needs future efforts to enhance the quality of adversarial audio synthesis.

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

# 7 APPENDIX

## 7.1 DATASET ANALYSIS

Figure 9 shows the length distribution of 1000 samples on Common Voice and Librispeech. Note that, to better sample the adversarial attack target, we filter out data points that have too small counts. From the selected 1000 samples, we generate our targeted attack dataset, viz., matching a target that has a similar length with the conditional text. Figures (10-11) illustrate the comparison of the text length between conditional text $y_o$ and target text $y_t$ on Common Voice and Librispeech, respectively. In general, the length of $y_o$ and $y_t$ are similar. Moreover, in both datasets, the text length has a huge variance, e.g., case 3 versus 30 in Librispeech. Compared Common Voice to Librispeech, the length of many cases on Librispeech is much larger than the maximum length (i.e., 100) on Common Voice.

## 7.2 ADDITIONAL ANALYSIS ON THE CONVERGENCE OF CTC LOSS AND LEVENSHTEIN DISTANCE AND LEARNING RATE DECAY

The CTC loss convergence processes across the 100 samples on Common Voice are shown in Figure 12. Moreover, we further analyse the convergence of Levenshtein distance $D_{ls}(\cdot)$ (Khare et al., 2019) in Figure 13, in order to depict how does our proposed SSA succeeds. Note that the Levenshtein distance is calculated as $D_{ls}(f(\mathcal{G}(z, y_o)), y_t)$, which indicates the distance between the transcription on the current synthesised audio $x$ and the target transcription $y_t$. We also analyse the dynamics of $\alpha$ in Figure 14.

From Figure 12, we can see that the CTC loss of most cases quickly converges to a value close to 0. The corresponding Levenshtein distance in Figure 13 exactly converges to 0 on these cases, suggesting successful attacks on most cases. For the cases that do not stop the optimization before reaching the maximum iteration step 8000, both the CTC loss and Levenshtein distance end with small values, which can be deemed as approximately successful attacks based on the related studies (Zhang et al., 2020) on adversarial attack in natural language processing. From Figure 14, generally different cases have their own learning rate schedule. On most cases, the learning rate needs to decay for at least 2 times. Only on some particular cases (e.g., case 2), directly using the initialized $\alpha$ without decay can find a successful adversarial example. This implies that our designed adaptive sign gradient descent algorithm enables the learning rate $\alpha$ to dynamically change according to the optimized loss.

## 7.3 ANALYSES ON THE AUDIO STYLE VECTOR $z$

We further compare the original audio style vector $z$ (i.e., sampled from the normal speech synthesis) and the adversarial $z$ (i.e., optimized by our SSA loss) on three cases as shown in Figure 8. For better visualization, we only plot partial of vector $z$. For instance, $z$ is reshaped from a $(192 \times 231)$ matrix with dimension determined by the conditional text in Figure 2, while we only plot $(2 \times 231)$ of them as shown in Figure 8 (a). In general, Figure 8 shows that the original $z$ and adversarial $z$ are significantly different. Namely, although both original and adversarial $z$ fluctuate around the mean 0 and share a comparable variance, their specific values for each dimension are quite different. This indicates that our SSA has more flexibility of searching for a successful attack. In contrast, the optimization space of previous audio dependent attacks is restricted around the original audio waveform by a norm bound as shown in Figure 7 (b).

## 7.4 ANALYSES OF ATTACK TRANSFER

Our SSA is designed to be ASR model dependent. In specific, the adversarial audios are synthesized based on Deep Speech. Therefore, as expected, these adversarial examples should pose limited threat to other ASR models. To validate such a hypothesis, we mount the successfully synthesised audio attacks on ESPnet (Watanabe et al., 2018) (i.e., an attention-based encoder-decoder network). In doing so, we randomly sample 30 successfully synthesised attacks (i.e., based on Deep Speech), input them to the ESPnet and calculate the levenstein distance ($LD$) with respect to the target text, where $LD = 0$ indicates a successful targeted attack. Results show that the success rate and $LD$ of

Table 3: The human speech recognition evaluations on the original and SSA synthesised audios.

| Type of audios | Human Translation WER |
|---|---|
| Before attack (original synthesised audios) | $18.52 \pm 7.46\%$ |
| After attack (SSA synthesised audios) | $22.30 \pm 5.05\%$ |

these transferred attacks on ESPnet are $0\%$ and $40.97 \pm 15.34$, respectively. This suggests that the adversarial audios generated by SSA are hardly transferable to a different ASR model.

## 7.5 HUMAN SPEECH RECOGNITION (SR) EVALUATION

The human speech recognition (SR) evaluation has been conducted. Specifically, we invite 5 participants to listen to 50 sample pairs, where each sample pair includes an original synthesised audio by CVAE and its corresponding synthesised adversarial audio optimized by our SSA. Each participant then writes down the corresponding translation text. The WER is calculated between the human translation and ground truth text. The averaged WER from the human evaluation is shown in Table 3, which indicates that the human translation performance is only slightly impacted compared to the original synthesised audios.

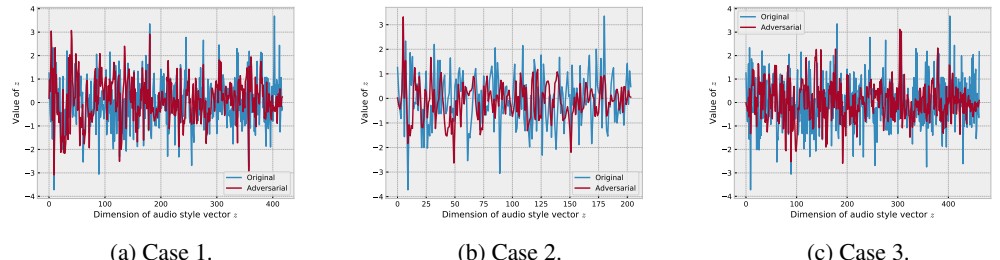

(a) Case 1.        (b) Case 2.        (c) Case 3.

Figure 8: Comparison of $z$ from the normal speech synthesis and the one optimized by our SSA loss.

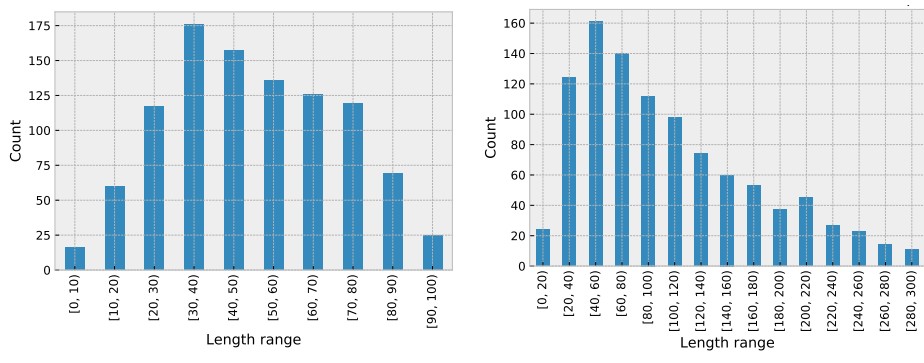

(a) Length distribution on Common Voice.

(b) Length distribution on Librispeech.

Figure 9: The length distribution of 1000 samples on Common Voice and Librispeech

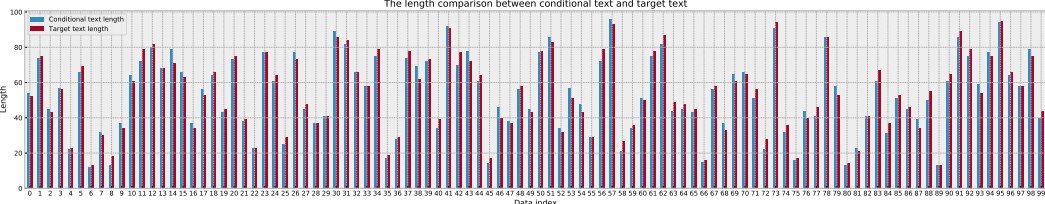

Figure 10: The length comparison between condition text $y_o$ and target text $y_t$ on Common Voice

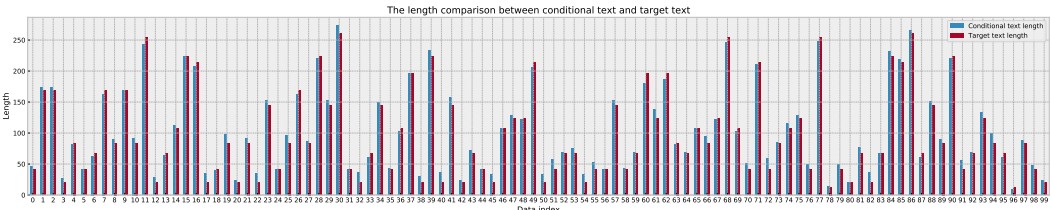

Figure 11: The length comparison between condition text $y_o$ and target text $y_t$ on Librispeech

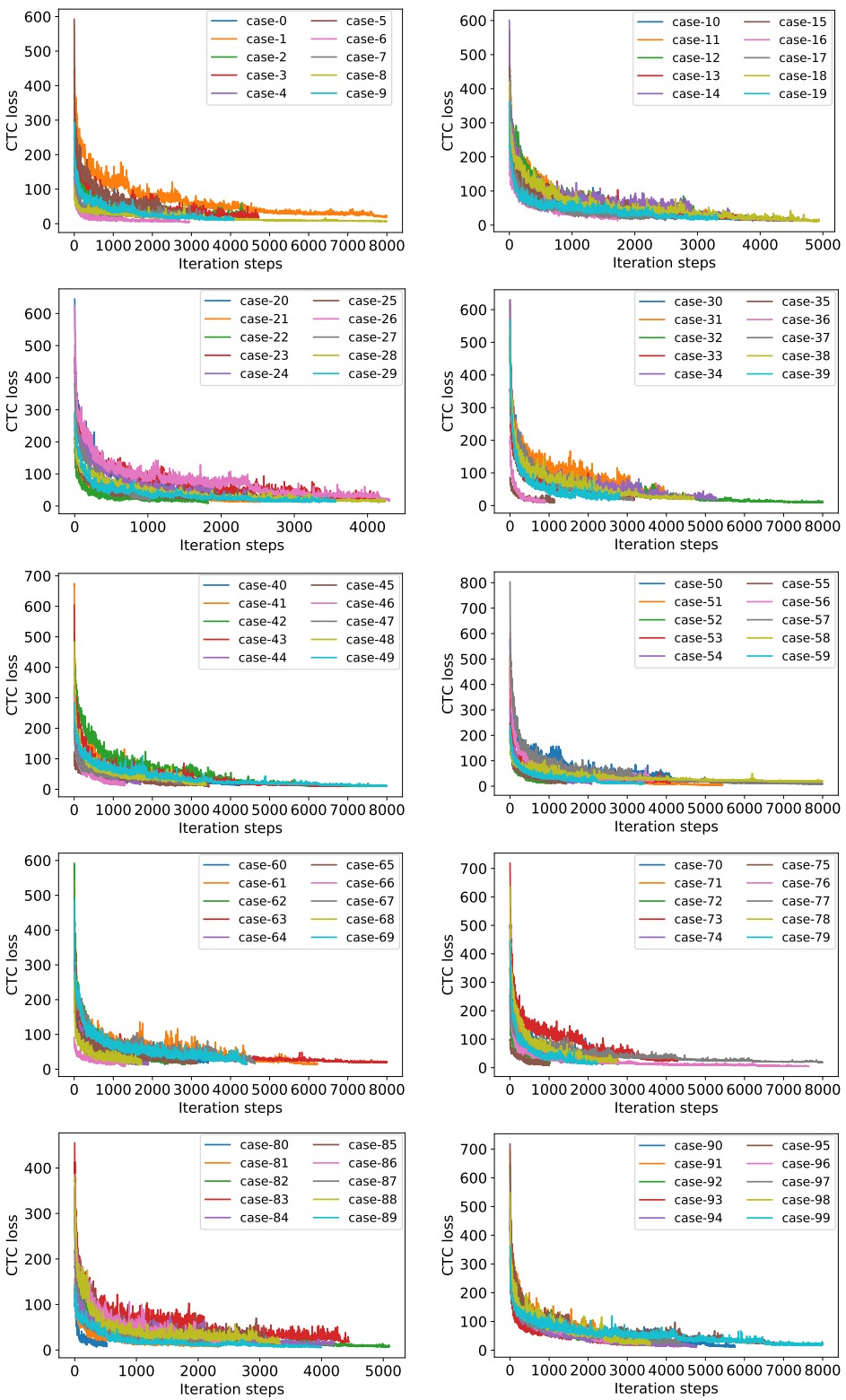

Figure 12: The CTC loss convergence of SSA on Common Voice.

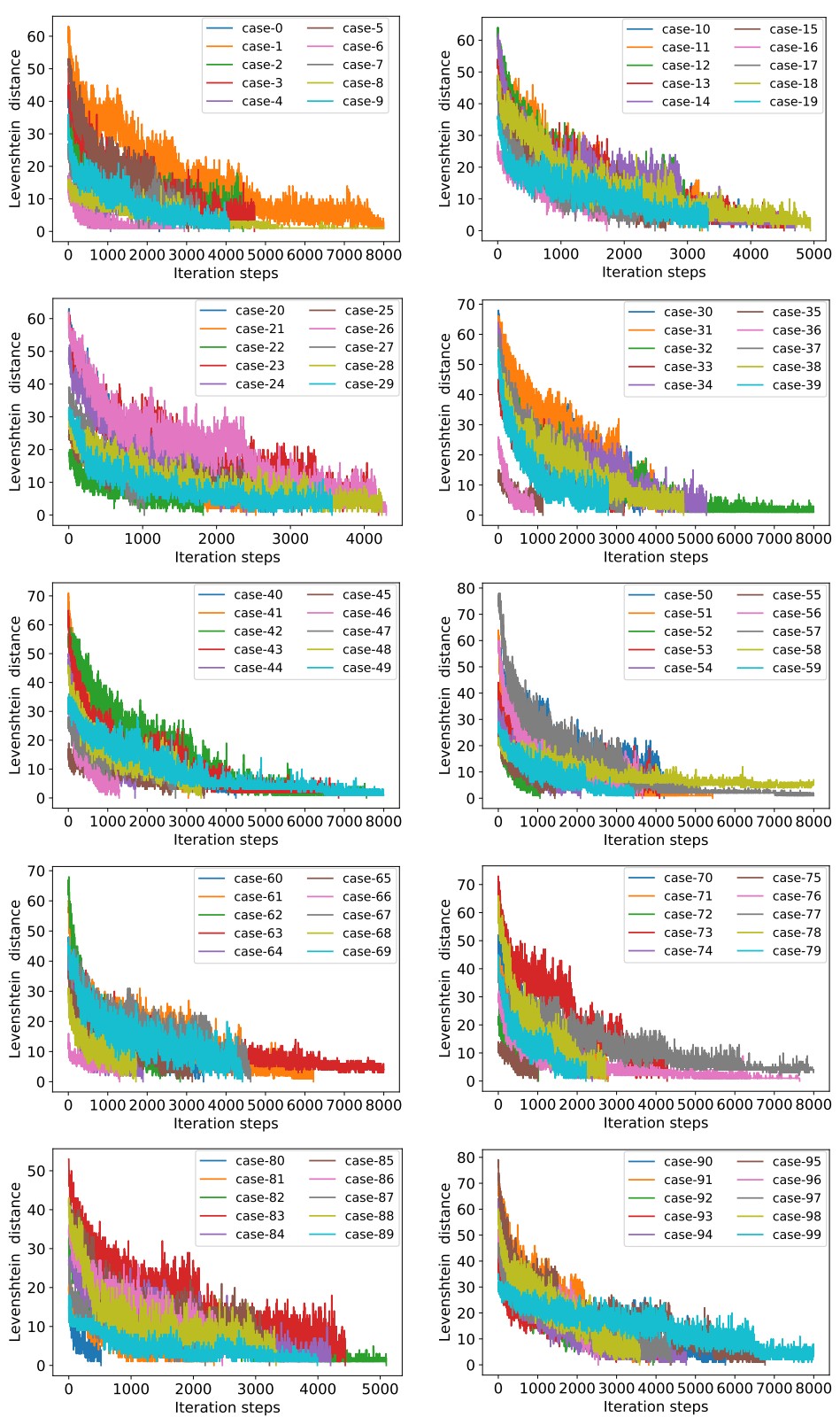

Figure 13: The Levenshtein distance convergence of SSA on Common Voice.

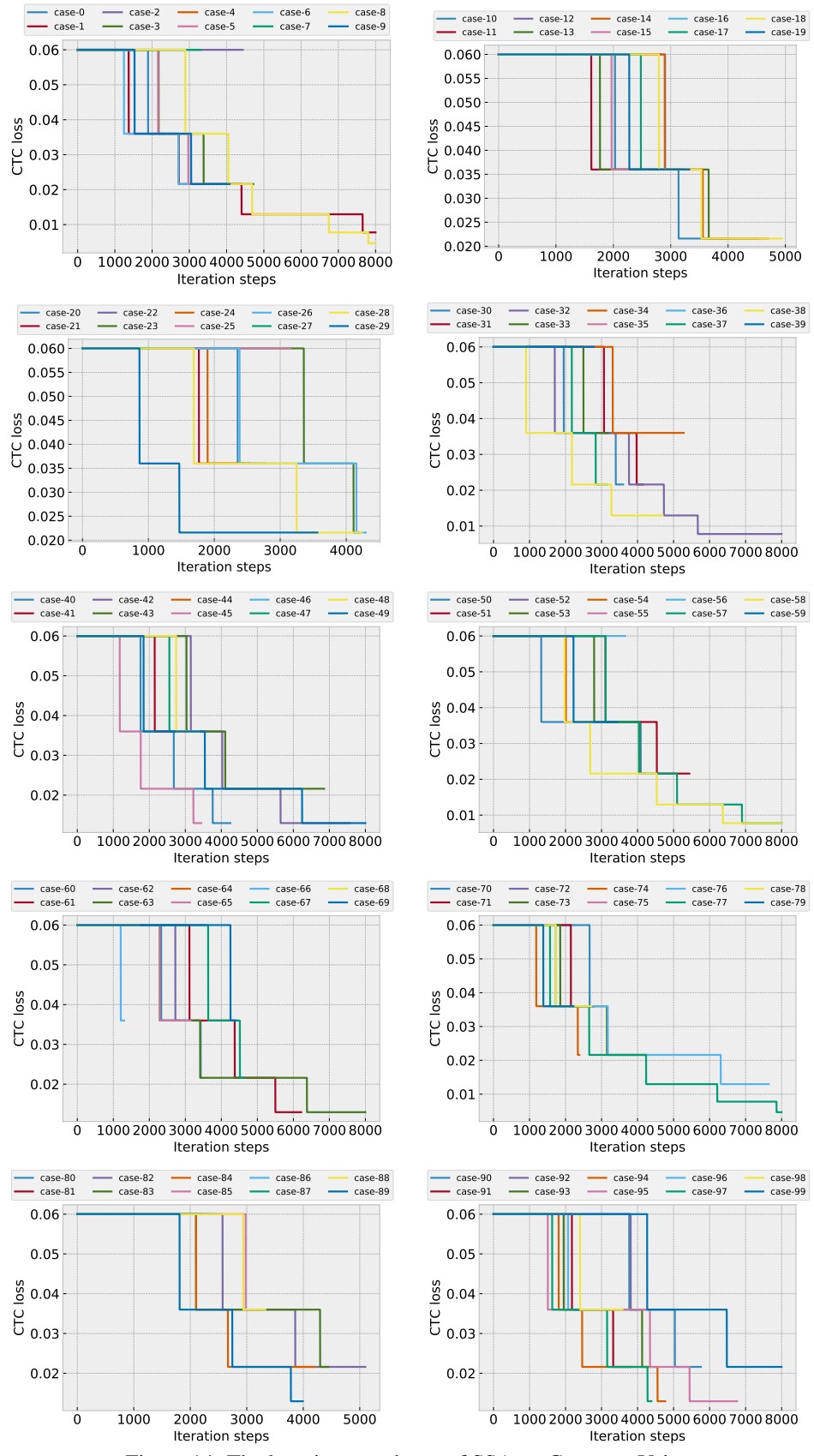

Figure 14: The learning rate decay of SSA on Common Voice.

