# OpenReview forum: "Synthesising Audio Adversarial Examples for Automatic Speech Recognition"
_ICLR.cc/2022/Conference — ICLR 2022 Submitted_

### Official Review · Reviewer_Krqb · 2021-11-02

**Correctness:** 4
**Technical Novelty And Significance:** 3
**Empirical Novelty And Significance:** 4
**Recommendation:** 8
**Confidence:** 4

**Main Review:**

The paper is well written, clear, the problem stated concisely, ample references
are provided, and the experiments are convincing. This novel type of attack -
synthesizing audio instead of perturbing - is interesting as one may not
have access to ground truth audio to perturb in an attack. Additionally, while
the overall approach is in theory complicated as it involves a STT and TTS system
and optimized for a specific loss, the authors present their method clearly.

The quantitative presentation is good and the inclusion of their adaptive sign
gradient decent algorithm may be useful for further researchers.

The authors show the validity of their SSA approach by demonstrating its superior
ability to attack an ASR system. Although, as the authors state, this is maybe
not so surprising given the fast that their SSA approach is able to modify
the input in many ways, whereas the perturbation approaches are limited in
how they can modify the input.

The only minor point is that there was no MOS done on samples the authors
generated. And, as the authors point out, and as this reviewer was able to listen,
some of the audio samples do not sound well. But, given the large scope of
the work, this can maybe be done in the future.

**Summary Of The Paper:**

In this work the authors explore a new vein of research in the area of audio
adversarial attacks wherein instead of focusing on perturbing existing audio,
they synthesize audio. To their best knowledge, and this reviewer, this is
the first such attempt. To this end, the authors use a conditional
variational auto-encoder as a speech synthesis (TTS) model and develop
a adaptive sign gradient decent algorithm in order to solve their SSA
optimization problem. Experiments are provided showing the effectiveness of
their approach, along with a detailed analysis of results.


**Summary Of The Review:**

Well written, novel contribution, good references, and compelling and complete experiments.

---

> ### Author Response · Authors · 2021-11-19
> **Response**
>
> We sincerely appreciate the reviewer's appreciation of our paper. The concern about the mean opinion score (MOS) evaluation on synthesised adversarial audios has been added. Results and the corresponding analyses are shown below.
>
> $\color{blue}\textbf{[Minor point] MOS evaluation was not done}$.
>
> Although the regularization loss $\mathcal{L}\_{reg}(z)$ in our SSA is designed to force our synthesised audios to be independently and identically distributed w.r.t. the audio generated in VITS~[1], we still find some synthesised audios not natural sounding enough. Therefore, we further evaluate the quality of synthesised adversarial audios by mean opinion score (MOS) tests. Specifically, we invited 20 participants to rate on 50 audio sample pairs, where each sample pair includes an original synthesised audio by CVAE and its corresponding synthesised adversarial audio optimized by our SSA.
> Each participant will listen to these 100 audio samples that are randomly shuffled. The naturalness rating score is scaled from 1 to 5. The MOS results are shown in the table below, where we can observe a slightly worse MOS score of our SSA synthesised audios (i.e., $3.39\pm0.29$) compared with that of original synthesised ones (i.e., $4.09\pm0.10$).
> This again indicates that there exists distortions in the synthesised audios by SSA.
> However, we further note that the difference of MOS scores between the two types of synthesised audios is not significant, which indicates that the distortions are still subjectively acceptable.
> Future works can focus on how to eliminate such distortions in the adversarial audio generation, such as redesigning the regularization loss $\mathcal{L}\_{reg}(z)$ in Eq (3).
>
> \begin{equation}\begin{array}{@{}|l|c|@{}}
> \\hline
>   \textrm{Type of audios} & \textrm{MOS} \\\\
> \\hline
>   \textrm{Before attack (original synthesised audios)} & 4.09\pm0.10 \\\\
> \\hline
>   \textrm{After attack (SSA synthesised audios)} & 3.39\pm0.29 \\\\
> \\hline
> \end{array}\end{equation}

---

> > ### Comment · Reviewer_Krqb · 2021-11-29
> > **MOS**
> >
> > Thank you for presenting an MOS test on the synthesized adversarial audios. This will make it clear to other readers quantitatively where the SAA stands.

---

### Official Review · Reviewer_Hbzp · 2021-11-02

**Correctness:** 3
**Technical Novelty And Significance:** 3
**Empirical Novelty And Significance:** 3
**Recommendation:** 6
**Confidence:** 4

**Main Review:**

Strength:

- The paper does provide a new adversarial attack mode that is different from the audio "dependent" attack (ADA) methods. While the ADA methods are usually performed by perturbing existing audio signals to the degree that human perception doesn't notice the difference, the proposed model can synthesize new speech signals that sound totally different from what they are intended to fool the ASR models with.

- After checking out some of the demo signals posted on the anonymized website, the reviewer found that the synthesized speech signals are indeed in good quality, i.e., they do sound natural and convey the original meaning, y_o.

- The targeted attack results are substantially better than the other models.

Weakness:

- It is not too clear to me what is the real-world attack scenario based on the proposed method. For example, I get the idea of ADA where an attacker secretly adds inaudible perturbation noise to the user's speech, so that the ASR system fails to recognize the voice command, e.g., for voice assistant services. However, what is the point of synthesizing new utterances from the attacker's point of view? The paper does not discuss this point of view, although it must nicely justify why the synthesis-based attacks are useful.

- There have been other adversarial models that are based on synthesized audio. For example, "hidden voice commands" [a], "inaudible voice commands" [b], and "DolphinAttack" [c] are the systems that are based on synthesized attack signals. While the latter two are to attack the system via inaudible sound (i.e., in the ultrasound frequency) and the hidden voice commands are basically noise signal which is hard to tell. I understand the difference between the proposed model and these existing models. But, the proposed model is lacking a discussion as to how the attacker can successfully use the synthesized voice, while the other afore-mentioned models are with convincing and clear attack scenarios. The paper might benefit from additional justification.
[a] https://people.eecs.berkeley.edu/~daw/papers/audio-dls18.pdf
[b] https://www.usenix.org/conference/nsdi18/presentation/roy
[c] https://dl.acm.org/doi/10.1145/3133956.3134052

- The authors claim that the generated attack signal sounds natural and can be perceived as the ground-truth label y_o. While the reviewer also observed that the samples uploaded on the website are convincing, I believe that a large-scale evaluation on (a) sound quality of the synthesized speech (b) human annotator's transcription is necessary. The authors indeed pointed out in the conclusion section that there are unnatural examples they observed.

- Related to the previous point, I believe that the loss function eq (3) isn't well justified. The first term is only to make sure that the produced speech signal is recognized as y_t, which is for the targeted attack part. However, the regularization part is just to make sure that the latent variables are following a standard normal distribution. Although I can see that the model is based on a CVAE voice synthesizer, and such regularization can indirectly enforce the model to still behave like a speech synthesizer as pre-trained. However, it is still not too clear if it is the best way.

- I am curious to know how the proposed model performs in the noisy/reverberant environment. Can it still fool the ASR system if the synthesized speech itself is contaminated by real-world acoustic perturbations?

- The proposed method is based on the white-box attack. Additional discussion on the black-box attack cases my strengthen the paper.

Other minor comments

- Definition 1 needs more elaboration. Hard to follow what the set is defined by due to the many intersection operations.


**Summary Of The Paper:**

The paper presents a new adversarial attack mode to ASR systems based on speech synthesis. The main idea is to synthesize speech utterances that naturally sound like ground-truth transcription y_o, but can fool an ASR system. It is a targeted attack model, where each attack signal has its own target label y_t. The paper also presents a CVAE-based speech synthesis system and a learning rate decay method for training.

**Summary Of The Review:**

I found this paper novel (to my best understanding) and interesting in that it provides a new adversarial attack mode. I wish that the paper is better justified as to when this kind of attack can be a real threat, instead of relying on the reader's own conjecture. The paper is missing discussions on the comparison to the other AIA methods. The demo signals sound natural, but it might be better to include some subjective test results as the optimization on the "naturality" part isn't too clear.

---

> ### Author Response · Authors · 2021-11-19
> **Response [Part 2/2]**
>
> $\color{blue}\textbf{[Q4] The regularization term in Eq (3)}$. The regularization in Eq (3) is designed to boost the audio style vector $z$ to be independently and identically distributed as its setting in CVAE training. In specific, the CVAE model in [1] is trained with sampling $z\in\mathcal{N}(\mathbf{0},\mathbf{1})$. Therefore, we regularize $z$ to be a normal distribution as shown by $\mathcal{L}_{reg}(z)$ in Eq (3). There may exist better regularization designs (e.g., a $p$ norm distance around the original $z$ from normally synthesised audios) that helps the optimization of our designed SSA. We will further explore them in our future work.
>
> $\color{blue} \textbf{[Q5] \\& [Q6] Consider more realistic settings}$. We will consider the suggested two settings, i.e., (1) the impact of noise from the real-world environment when playing the adversarial audio through the air and (2) the black-box assumption of ASR model, in our future work. These settings will help on a more realistic speech synthesis based attack by considering the situation in the wild.
>
> $\color{blue}\textbf{Minor comment: Definition 1 needs more elaboration}$.
>
> Inspired by the comment, we have added under brackets to indicate different conditions, thus making the formula easier to follow. The revised Definition 1 \& 2 can refer to the updated manuscript.
>
> [a] https://people.eecs.berkeley.edu/~daw/papers/audio-dls18.pdf
>
> [b] https://www.usenix.org/conference/nsdi18/presentation/roy
>
> [c] https://dl.acm.org/doi/10.1145/3133956.3134052
>
> [d] Kim et al. Conditional Variational Autoencoder with Adversarial Learning for End-to-End Text-to-Speech. ICML 2021.

---

> ### Author Response · Authors · 2021-11-19
> **Response [Part 1/2]**
>
> We thank the reviewer's comments and suggestions. The summaries of reviewer's general concerns and the corresponding responses are shown below.
>
> $\color{blue}\textbf{[Q1] The real-world scenario of our SSA}$.
>
> Our SSA is proposed as a new routine for generating audio adversarial examples through speech synthesis, which provides more choices for attacking ASR systems besides
> previous audio perturbation based attacks.
> There are plenty of scenarios that our speech synthesising based attack (SSA) could apply to.
> For instance, in the smart home, there are multiple voice assistant systems existing together.
> Our SSA can hijack a weather speaker to synthesis an audio that sound to be with a benign semantic content, e.g., "it is going to rain later today". Given the targeted attack setting of SSA, this synthesised audio can sound natural to the house owner but mislead another financial voice assistant to commit an illegal command based on its translation, e.g.,"transfer one thousand dollars to the account XXX".
>
>
> $\color{blue}\textbf{[Q2] More literature discussions}$.
>
> Thanks the reviewer for suggesting the related references, which have been added in the revised manuscript accordingly.
> In specific, [a] studies an audio attack that starts from non-speech (e.g., a piece of classic music), while existing audios are stilled required to mount perturbations on. [b] and [c] propose to modulate voice commands on ultrasonic carriers (e.g., frequency > 20 kHz) to achieve inaudibility, where the scenario is an adversary that stands on the road and silently controls the voice command assistant systems. This therefore is different from our scope of generating natural sounding adversarial audios. In addition, the attack scenario for our SSA has been discussed in the response to [Q1].
>
> $\color{blue}\textbf{[Q3] Evaluations on synthesised audio quality and human annotator's transcription}$.
>
>
> $\textbf{Audio quality evaluation based on the mean opinion score (MOS)}$
>
> The audio quality evaluation based on mean opinion score (MOS) tests and corresponding analyses are shown below.
>
> Although the regularization loss $\mathcal{L}\_{reg}(z)$ in our SSA is designed to force our synthesised audios to be independently and identically distributed w.r.t. the audio generated in VITS~[1], we still find some synthesised audios not natural sounding enough. Therefore, we further evaluate the quality of synthesised adversarial audios by mean opinion score (MOS) tests. Specifically, we invited 20 participants to rate on 50 audio sample pairs, where each sample pair includes an original synthesised audio by CVAE and its corresponding synthesised adversarial audio optimized by our SSA.
> Each participant will listen to these 100 audio samples that are randomly shuffled. The naturalness rating score is scaled from 1 to 5. The MOS results are shown in the table below, where we can observe a slightly worse MOS score of our SSA synthesised audios (i.e., $3.39\pm0.29$) compared with that of original synthesised ones (i.e., $4.09\pm0.10$).
> This again indicates that there exists distortions in the synthesised audios by SSA.
> However, we further note that the difference of MOS scores between the two types of synthesised audios is not significant, which indicates that the distortions are still subjectively acceptable.
> Future works can focus on how to eliminate such distortions in the adversarial audio generation, such as redesigning the regularization loss $\mathcal{L}\_{reg}(z)$ in Eq (3).
> \begin{equation}\begin{array}{@{}|l|c|@{}}
> \\hline
>   \textrm{Type of audios} & \textrm{MOS} \\\\
> \\hline
>   \textrm{Before attack (original synthesised audios)} & 4.09\pm0.10 \\\\
> \\hline
>   \textrm{After attack (SSA synthesised audios)} & 3.39\pm0.29 \\\\
> \\hline
> \end{array}\end{equation}
>
> $\textbf{Human annotator's transcription}$
>
> The human speech recognition (SR) evaluation has been added. Specifically, we invite 5 participants to listen to 50 sample pairs, where each sample pair includes an original synthesised audio by CVAE and its corresponding synthesised adversarial audio optimized by our SSA. Each participant then writes down the corresponding translation text. The WER is calculated between the human translation and ground truth text.
> The averaged WER from the human evaluation is shown in the table below, which indicates that the human translation performance is only slightly impacted compared to the original synthesised audios.
>
> \begin{equation}\begin{array}{@{}|l|c|@{}}
> \\hline
>   \textrm{Type of audios} & \textrm{Human Translation WER} \\\\
> \\hline
>   \textrm{Before attack (original synthesised audios)} & 18.52\pm7.46\\% \\\\
> \\hline
>   \textrm{After attack (SSA synthesised audios)} & 22.30\pm5.05\\%  \\\\
> \\hline
> \end{array}\end{equation}

---

### Official Review · Reviewer_4HAP · 2021-11-03

**Correctness:** 3
**Technical Novelty And Significance:** 3
**Empirical Novelty And Significance:** 2
**Recommendation:** 5
**Confidence:** 4

**Main Review:**

The most interesting component of this paper is the fact that adversarial attacks can be constructed by manipulation of the prosodic characteristics of synthetic speech.

However, there are a number of unanswered questions about this adversarial attack.
First and foremost, it is not at all clear that the use of synthesis and prosodic manipulation is a more compelling attack than adding adversarial noise to carrier speech.  The context where there is no available carrier speech is unconvincing -- an attacker can record or find some short speech transcript as easily (if not much easier) than they can train a TTS model that can be updated by CVAE.

The white-box assumption is reasonable for this kind of proof of concept, but it's less convincing than a black-box attack.

There is no assessment of TTS quality either of the un-modified initial utterance (that should be recognized correctly by the target ASR model) or the adversarial utterance.  The adversarial examples are fairly intelligible as the original phrase, but the naturalness is substantially impaired (i.e. to my ear many are obviously synthetic speech).  This is a limitation especially in comparison to audio dependent attacks (ADA) which sound like natural speech with a small amount of background noise.  For this attack to be completely successful the synthetic audio should be indistinguishable from human speech.  This isn't assessed.

Smaller questions:
* Section 1: Please elaborate on the use of Google Assistant and Apple Siri that are 'safety-critical'
* Section 4.1: In what case is 'benign audio unavailable'
* Section 4.1: the claim that this objective function "guarantees [that] the audio x [is] natural sounding" is over stated.  TTS aims for natural sounding speech, but this is far from guaranteed.
* Section 4.2: why is dependency on some ground truth text preferable to dependency on some ground truth audio?
* Section 4.2: what is the rationale for including levenstein distance as distinct from WER?
* Section 5.1: there is no evaluation of Synthesized Audio Quality



**Summary Of The Paper:**

This paper describes a white-box attack on ASR systems using TTS.  The main technique is (somewhat) standard adversarial training with the constraint that the CVAE component of a TTS model is the only component that is updated.  Since this component is responsible for prosodic and other non-segmental qualities of speech synthesis, the expectation is that the naturalness of the synthesis will not be impacted, but the ASR recognition will be modified.

**Summary Of The Review:**

Technically this paper is sound. However, the contribution here is rather narrow.  It is interesting, but not especially consequential, that CVAE manipulation can be used to localize the contribution of an adversarial attack.  It is unclear that the presence or absence of a carrier speech signal in favor of a carrier tts utterance is a consequential modification of this attack.

---

> ### Author Response · Authors · 2021-11-19
> **Response [Part 2/2]**
>
> Small questions are answered as below.
>
> $\color{blue}\textbf{[Q4.1] Section 1}$.
>
> The claim has been revised as: This paper mainly focuses on the ASR domain, where many voice assistant systems could be hijacked or controlled by the audio adversarial examples constructed by an attacker.
>
> $\color{blue}\textbf{[Q4.2] Section 4.1 (1)}$.
>
> One scenario of benign audio unavailable could be the case when a user of a voice assistant system (VAS) does not give any command. For instance, the user stays at home yet saying nothing to the VAS. Thus no benign audio is available to add perturbations on.
>
> $\color{blue}\textbf{[Q4.3] Section 4.1 (2)}$.
>
> As suggested by the reviewer, we have revised the statement in Definition 2 as `where $o(x)=y_o$ indicates that the synthesised audio $x$ correctly conveys its semantic content $y_o$.'
>
>
> $\color{blue}\textbf{[Q4.4] Section 4.2 (1) Dependency on ground truth text or audio?}$.
>
> Depending on ground truth text may enable more patterns of adversarial audios, since the synthesised audios could be with different speakers, speaking speed and emotion styles. In contrast, the ground truth audio dependent attack can only add perturbations within a restricted neighborhood around the ground truth audio.
> Moreover, our aim of proposing audio independent attack is to provide a new routine of generating audio adversarial examples besides previous waveform perturbation based attacks.
>
> $\color{blue}\textbf{[Q4.5] Section 4.2 (2)}$.
>
> Besides the word error rate (WER) introduced to compare with previous reported results, we further involve the levenstein distance ($LD$) to show the character level difference between the predicted transcription and the targeted attack text.
> In our experiments, a successful attack is defined based on $LD=0$.
>
> $\color{blue}\textbf{[Q4.6] Section 5.1}$.
>
> We have added the mean opinion score (MOS) tests, which can refer to the response to \textbf{[Q3] Assessment of TTS quality}.
>
> References.
>
> [1] Kim et al. Conditional Variational Autoencoder with Adversarial Learning for End-to-End Text-to-Speech. ICML 2021.

---

> ### Author Response · Authors · 2021-11-19
> **Response [Part 1/2]**
>
> We thank the reviewer's comments and suggestions. The summary of reviewer's major concerns and our detailed responses are shown below.
>
> $\color{blue}\textbf{[Q1] The setting of no available carrier speech is unconvincing}$.
>
> Our SSA is proposed as a new routine for generating audio adversarial examples through speech synthesis, which provides more choices for attacking ASR systems besides
> previous audio perturbation based attacks.
> There are plenty of scenarios that our speech synthesising based attack (SSA) could apply to.
> For instance, in the smart home, there are multiple voice assistant systems existing together.
> Our SSA can hijack a weather speaker to synthesis an audio that sound to be with a benign semantic content, e.g., "it is going to rain later today". Given the targeted attack setting of SSA, this synthesised audio can sound natural to the house owner but mislead another financial voice assistant to commit an illegal command based on its translation, e.g.,"transfer one thousand dollars to the account XXX".
>
>
> $\color{blue}\textbf{[Q2] Black-box attack}$.
>
> We will consider the more realistic yet challenging black-box setting in our future work. We also hope this study can inspire other works on exploring more realistic settings (e.g., playing over the air, transferability of attacks across multiple ASR models) under the SSA framework.
>
> $\color{blue}\textbf{[Q3] Assessment of TTS quality}$.
>
> The assessment of TTS quality and corresponding analyses have been added as shown below.
>
> Although the regularization loss $\mathcal{L}\_{reg}(z)$ in our SSA is designed to force our synthesised audios to be independently and identically distributed w.r.t. the audio generated in VITS~[1], we still find some synthesised audios not natural sounding enough. Therefore, we further evaluate the quality of synthesised adversarial audios by mean opinion score (MOS) tests. Specifically, we invited 20 participants to rate on 50 audio sample pairs, where each sample pair includes an original synthesised audio by CVAE and its corresponding synthesised adversarial audio optimized by our SSA.
> Each participant will listen to these 100 audio samples that are randomly shuffled. The naturalness rating score is scaled from 1 to 5. The MOS results are shown in the table below, where we can observe a slightly worse MOS score of our SSA synthesised audios (i.e., $3.39\pm0.29$) compared with that of original synthesised ones (i.e., $4.09\pm0.10$).
> This again indicates that there exists distortions in the synthesised audios by SSA.
> However, we further note that the difference of MOS scores between the two types of synthesised audios is not significant, which indicates that the distortions are still subjectively acceptable.
> Future works can focus on how to eliminate such distortions in the adversarial audio generation, such as redesigning the regularization loss $\mathcal{L}\_{reg}(z)$ in Eq (3).
>
> \begin{equation}\begin{array}{@{}|l|c|@{}}
> \\hline
>   \textrm{Type of audios} & \textrm{MOS} \\\\
> \\hline
>   \textrm{Before attack (original synthesised audios)} & 4.09\pm0.10 \\\\
> \\hline
>   \textrm{After attack (SSA synthesised audios)} & 3.39\pm0.29 \\\\
> \\hline
> \end{array}\end{equation}

---

> > ### Comment · Reviewer_4HAP · 2021-11-22
> > **response to response**
> >
> > Thank you for including the Assessment of TTS quality in the manuscript.  This definitely strengthens the paper.
> >
> > > However, we further note that the difference of MOS scores between the two types of synthesised audios is not significant, which indicates that the distortions are still subjectively acceptable.
> > By what analysis is 3.39 +- 0.29 not significantly worse than 4.09 +- 0.10?  For TTS systems, in broad strokes, a score about 4 is considered very natural, while a score in the 3-3.5 range is clearly identifiable as synthetic, but reasonably acceptable.
> >
> > Regarding the attack setting -- I still don't understand where it would be possible to synthesize speech, but not play audio.  The only change between the text based attack and the audio based attack is that in the text based attack, the audio must first be generated by TTS.  It is not clear that the CVAE manipulation is preferable to traditional audio based attacks that could be, say, based on synthetic audio input rather than "real" audio inputs or based on some found audio that is manipulated and replayed.
> >
> > I truly thank the authors for their work in improving the paper.  However, I stand by my original summary -- the work is technically interesting, but the contribution is very narrow. I don't find the distinction between (real) audio dependence and (synthetic) audio dependence to be significant enough to support ICLR publication.

---

> > > ### Author Response · Authors · 2021-11-23
> > > **Response to further comments**
> > >
> > > We sincerely thank the reviewer's further comments and great efforts in assessing our paper. The responses are shown below.
> > >
> > > $\color{blue}\textbf{[Q1] Assessment of TTS quality.}$
> > >
> > > Our MOS guidelines [1] utilised are shown below.
> > >
> > > 1. Very bad: $\textit{Incomprehensible}$, only tiny pieces and fragments can be understood; strange and un-humanlike.
> > >
> > > 2. Bad: $\textit{Unclear}$; speaker's accent is annoying and uncomfortable.
> > >
> > > 3. Fair: $\textit{Acceptable}$ despite minor mistakes in pronunciation and prosody; understandable with a slight effort.
> > >
> > > 4. Good: $\textit{Near-perfection}$; clear, understandable and pleasing to the ear, though subtle imperfections can still be perceived.
> > >
> > > 5. Excellent: $\textit{Perfect}$, in line with the audio broadcasting standards; words flow with very natural intonation.
> > >
> > > We admit $3.39\pm0.29$ is not as good as $4.09\pm0.10$ (i.e., approximately deemed as Near-perfection), but it is between Acceptable and Near-perfection. Thus we say the difference is not significant from the qualitative perspective. In addition, $3.39\pm0.29$ is comparable to some TTS baselines, such as $3.49$ in [2]. In summary, we believe the distortions from our SSA are subjectively acceptable. Given our SSA as the first work in this direction, we believe the ongoing advancement of TTS and future improved regularization of speech naturalness could boost SSA to pose increasing threat to the security of ASR systems.
> > >
> > > $\color{blue}\textbf{[Q2] Where is SSA applicable?}$.
> > >
> > > In general, our SSA can pose threats in any scenario where the audios are generated by TTS model and used in downstream ASR tasks. For instance, in the tasks of automatic books/news reading [3] and synthesised audio presentations for videos, the text content is converted to audios for either learning (e.g., listening to audiobooks) or entertainment (e.g., creating short videos). Such synthesised audios are thereby used for downstream ASR tasks, such as auto caption in Youtube or TikTok. Herein, our SSA can be applied to synthesise adversarial audios that are naturally sounded but output harmful translated captions containing sensitive/immoral information.
> > >
> > > $\color{blue}\textbf{[Q3] Distinction between \textit{real} and \textit{synthetic} audio dependence.}$
> > >
> > > The distinctions between the real audio dependence (i.e., audio perturbation based attack) and the synthetic audio dependence (i.e., speech synthesis based attack) can be summarised into two folds.
> > >
> > > $\textbf{First}$, synthetic audio dependence is more unrestricted, viz., the synthesised adversarial audios could be with plenty of rhythms, from different speakers, and in different speed. In contrast, the audio perturbation based attack is restricted to search within a norm bounded neighbourhood around the real audio. Such unrestricted attacks (e.g., [4]) using generative model have been widely investigated in computer vision but seldom studied in the speech domain.
> > >
> > > $\textbf{Second}$, synthetic audio dependence does not need carrier audios. Specifically, in the setting of real audio dependence, an existing audio is required as a carrier. Such carrier audio however may not be always available. Moreover, even when carrier audios exist, they may not cover every possible sentence. In some cases, the adversarial audio is expected to convey specific semantic content. In contrast, synthetic audio dependence could use the TTS model to synthesise any semantic content in different styles through simply adjusting the conditional text (as shown in Figure 2).
> > >
> > > [1] https://en.wikipedia.org/wiki/Mean_opinion_score
> > >
> > > [2] Zen et al. Fast, compact, and high quality LSTM-RNN based statistical parametric speech synthesizers for mobile devices.Interspeech 2016.
> > >
> > > [3] https://www.audiobooks.com/
> > >
> > > [4] Song et al. Constructing unrestricted adversarial examples with generative models. NeurIPS 2018.

---

### Official Review · Reviewer_aMq5 · 2021-11-06

**Correctness:** 3
**Technical Novelty And Significance:** 2
**Empirical Novelty And Significance:** 3
**Recommendation:** 6
**Confidence:** 4

**Main Review:**

(1) The main contribution of this work is to extend the previous work by replacing audio inputs with the style vector z in the CVAE speech synthesizer. The framework is new, and the results are promising. However, the scientific depth seems a bit shallow. Since ICLR is a top machine learning conference, I would suggest the authors further highlight the major contributions, especially the scientific novelty, of this work and potential impacts to related research fields.
(2) Additional experiments should be conducted to confirm the effectiveness of the proposed approach. In previous works, such as (Carlini & Wagner, 2018), an audio input is given, and the goal is to modify the input so that human listeners cannot perceive the modifications while ASR performance is considerably affected. On the other hand, the SSA framework directly generates audio samples from the text. Thus, an additional experiment is required: using another ASR system, trained from other datasets or constructed by another model architectures (such as LAS or hybrid CTC/Attention), to recognize the adversarial audio samples synthesized by SSA. We expected to see that the adversarial audio samples can only attack the target ASR system while not affecting the other ASR systems.
(3) The authors state that “Note that the audio x generated by CVAE model G(.) has been tested to be natural sounding using the mean opinion score obtained from Amazon Mechanical Turk” Please show the results of listening tests in the paper.
(4) Since the goal of this work is to attack ASR systems, it is meaningful to show that human recognition performance is not affected. Please conduct additional experiments and include the results in the paper.
(5) Figure 7 compares different adversarial samples. Since the samples generated by (Carlini & Wagner, 2018) and SSA are from different sources, it is reasonable that the samples show very different patterns. To us, the comparison does not provide useful information. It would be more meaningful to show and compare the vector z trained from the loss of normal speech synthesis and the SSA loss.
(6) It is very nice that the authors provide demo samples on a website. By listening to the samples provided on the website, however, we note that the samples generated by SSA present perceivable distortions. The quality is worse than the ones of (Carlini & Wagner, 2018). I am not sure whether the distortions come from the speech synthesizer or SSA loss. Please comment on this or provide additional experimental results using another speech synthesizer.

**Summary Of The Paper:**

This paper presents a speech synthesising-based attack (SSA) framework that generates audio examples entirely from scratch to attack automatic speech recognition (ASR) systems. A conditional variational auto-encoder (CVAE) based speech synthesiser is trained based on a combined connectionist temporal classification (CTC) and regularization loss. As compared to existing methods, the proposed approach serves as the first work that generates adversarial samples without audio inputs. The results confirm that the proposed SSA framework can attack the ASR system successfully. A comprehensive analysis of different hyper-parameters to the achievable performance is provided. Moreover, demo audio samples are provided by a website link.


**Summary Of The Review:**

The promising results confirm the effectiveness of the proposed SSA to attack the ASR system. The authors also provide a comprehensive analysis of hyper-parameters setups to the achievable performance. Moreover, demo audio samples are provided by a website link. The paper is well-written, and the theoretical part should be correct. However, we think additional experiments should be conducted before the paper can be accepted for publication. Moreover, the authors are suggested to summarize the major theoretical contributions on this study and discuss potential impacts to related research fields

---

> ### Author Response · Authors · 2021-11-19
> **Response [Part 2/2]**
>
> $\color{blue}\textbf{[Q5] Compare vector $z$ before and after attack}$.
>
> We further compare the original audio style vector $z$ (i.e., sampled from the normal speech synthesis) and the adversarial $z$ (i.e., optimized by our SSA loss) on three cases as shown in Figure 8 in the updated manuscript. For better visualization, we only plot partial of vector $z$. For instance, $z$ is reshaped from a $(192\times 231)$ matrix with dimension determined by the conditional text in Figure 2, while we only plot $(2\times 231)$ of them as shown in Figure 8 (a).
> In general, Figure 8 shows that the original $z$ and adversarial $z$ are significantly different. Namely, although both original and adversarial $z$ fluctuate around the mean $0$ and share a comparable variance, their specific values for each dimension are quite different. This indicates that our SSA has more flexibility of searching for a successful attack. In contrast, the optimization space of previous audio dependent attacks is restricted around the original audio waveform by a norm bound as shown in Figure 7(b). The added analyses can refer to Appendix 7.3 in the updated manuscript.
>
> $\color{blue}\textbf{[Q6] Quality evaluation of synthesised adversarial audios}$.
>
> As mentioned in the manuscript, there is indeed distortions in synthesised adversarial audios. Such distortions may be caused by the challenging optimization of SSA back-propogated through the concatenation of the ASR and TTS models. Although $z$ is regularized to be independently and identically distributed with its training setting in CVAE [2], it still ends with some distortions.
> To further evaluate the quality towards how natural the synthesised adversarial audios are sounded, we have added the mean opinion score (MOS) tests.
>
> Specifically, we invited 20 participants to rate on 50 sample pairs, where each sample pair includes an original synthesised audio by CVAE and its corresponding synthesised adversarial audio optimized by our SSA.
> Each participant will listen to these 100 audio samples that are randomly shuffled. The naturalness rating score is scaled from 1 to 5. The MOS results are shown in the table below, where we can observe a slightly worse MOS score of our SSA synthesised audios (i.e., $3.39\pm0.29$) compared with that of original synthesised ones (i.e., $4.09\pm0.10$).
> This again indicates that there exists distortions in the synthesised audios by SSA.
> However, we further note that the difference of MOS scores between the two types of synthesised audios is not significant, which indicates that the distortions are still subjectively acceptable.
> Future works can focus on how to eliminate such distortions in the adversarial audio generation, such as redesigning the regularization loss $\mathcal{L}_{reg}(z)$ in Eq (3).
>
> \begin{equation}\begin{array}{@{}|l|c|@{}}
> \\hline
>   \textrm{Type of audios} & \textrm{MOS} \\\\
> \\hline
>   \textrm{Before attack (Original synthesised audios)} & 4.09\pm0.10 \\\\
> \\hline
>   \textrm{After attack (SSA synthesised audios)} & 3.39\pm0.29 \\\\
> \\hline
> \end{array}\end{equation}
>
> References.
>
> [1] Shinji et al. ESPnet: End-to-End Speech Processing Toolkit. Interspeech 2018.
>
> [2] Kim et al. Conditional Variational Autoencoder with Adversarial Learning for End-to-End Text-to-Speech. ICML 2021.

---

> ### Author Response · Authors · 2021-11-19
> **Response [Part 1/2]**
>
> We thank the reviewer for assessing our paper and raising insightful comments towards improving our paper. Our responses are shown below point-by-point.
>
> $\color{blue}\textbf{[Q1] Highlight the major contributions}$.
>
> Our main contribution is that we provide a new routine for generating audio adversarial examples through speech synthesis. This sheds light on a more general principle of adversarial attacks, viz., $\textit{\textbf{any}}$ audio that deceives ASR models yet fails to deceive human beings would cause security issues in speech recognition. Particularly, in previous audio attacks, an existing audio is usually required as a carrier to add perturbations on it. This makes the adversarial audio being explored in a restricted neighborhood around the carrier waveform. Moreover, the carrier speech may not be available or accessible in some cases. In contrast, our speech synthesis based attack (SSA) could construct adversarial audio from scratch, which thus servers as a new threat model on automatic speech recognition (ASR) based systems. To effectively solve the SSA problem, we further design an adaptive sign gradient optimization algorithm. Given the ongoing advancement of text-to-speech models, our SSA will pose increasing threat to the security of ASR systems.
>
> $\color{blue}\textbf{[Q2] Transfer the generated adversarial audios to another ASR model}$.
>
> Our SSA is designed to be ASR model dependent. In specific, the adversarial audios are synthesized based on Deep Speech. Therefore, as expected, these adversarial examples should pose limited threat to other ASR models. To validate such a hypothesis, we mount the successfully synthesised audio attacks on ESPnet [1] (i.e., an attention-based encoder-decoder network). In doing so, we randomly sample 30 successfully synthesised attacks (i.e., based on Deep Speech), input them to the ESPnet and calculate the levenstein distance ($LD$) with respect to the target text, where $LD=0$ indicates a successful targeted attack. Results show that the success rate and $LD$ of these transferred attacks on ESPnet are $0\%$ and $40.97\pm15.34$, respectively. This suggests that the adversarial audios generated by SSA are hardly transferable to a different ASR model.
>
> $\color{blue}\textbf{[Q3] Show listening results}$.
>
> First, this sentence aims to clarify that CVAE $\textbf{in reference [3]}$ has been verified to synthesis naturally sounded audios via the mean opinion score (MOS) tests from Amazon Mechanical Turk. We have revised the manuscript accordingly to reduce possible confusion. Second, our own MOS tests towards the quality of synthesised adversarial audios has also been added as shown in the response for Q6.
>
> $\color{blue}\textbf{[Q4] Add human recognition performance evaluation}$.
>
> The human speech recognition (SR) evaluation has been added. Specifically, we invite 5 participants to listen to 50 sample pairs, where each sample pair includes an original synthesised audio by CVAE and its corresponding synthesised adversarial audio optimized by our SSA. Each participant then writes down the corresponding translation text. The WER is calculated between the human translation and ground truth text. The averaged WER from the human evaluation is shown in the table below, which indicates that the human translation performance is only slightly impacted compared to the original synthesised audios.
>
> \begin{equation}\begin{array}{@{}|l|c|@{}}
> \\hline
>   \textrm{Type of audios} & \textrm{Human Translation WER} \\\\
> \\hline
>   \textrm{Before attack (original synthesised audios)} & 18.52\pm7.46\\% \\\\
> \\hline
>   \textrm{After attack (SSA synthesised audios)} & 22.30\pm5.05\\%  \\\\
> \\hline
> \end{array}\end{equation}

---

### Decision · Program_Chairs · 2022-01-20

**Decision:**

Reject

**Comment:**

This paper introduces a way of generating adversarial speech samples to attack an ASR system based on speech synthesis.  The proposed approach, by the name of Speech Synthesizing based Attack (SSA) , does not rely on real speech to create adversasial samples but rather create them purely from text using  a conditional variational auto-encoder.  Experiments are conducted on three datasets and the results appear to support the effectiveness of the proposed approach.

All reviewers consider the paper technically sound and well written. Overall, the work is interesting. It pursues another possibility for a threat model on ASR and may inspire related work in the community. Reviewers also raised a number of concerns most of which have been cleared by the authors' rebuttal.  However, there are two major ones standing. One is the perceptual quality of the synthesized speech and the other is the justification of the application scenarios in the real world.  In the follow-up MOS experiments, there seems to be a noticeable difference (4.09 vs 3.39) which means the synthesized speech does sacrifice quality.  From the real-world application perspective, the scenarios proposed by the authors seem to be a bit contrived.  These two drawbacks are considered significant and need further investigation and justification.